# Behavior of Correlation Functions in the Dynamics of the Multiparticle Quantum Arnol’d Cat

**DOI:** 10.3390/e26070572

**Published:** 2024-06-30

**Authors:** Giorgio Mantica

**Affiliations:** 1Center for Non-Linear and Complex Systems, Università dell’Insubria, Via Valleggio 11, 22100 Como, Italy; giorgio.mantica@uninsubria.it; 2Istituto Nazionale di Alta Matematica “F. Severi”, GNFM Gruppo Nazionale per la Fisica Matematica, P. le Aldo Moro 5, 00185 Rome, Italy; 3I.N.F.N. Gruppo Collegato di Como, Sezione di Milano, Via Celoria 16, 20133 Milan, Italy

**Keywords:** classical and quantum autocorrelation functions, chaos, quantum-to-classical transition

## Abstract

The multi-particle Arnol’d cat is a generalization of the Hamiltonian system, both classical and quantum, whose period evolution operator is the renowned map that bears its name. It is obtained following the Joos–Zeh prescription for decoherence by adding a number of scattering particles in the configuration space of the cat. Quantization follows swiftly if the Hamiltonian approach, rather than the semiclassical approach, is adopted. The author has studied this system in a series of previous works, focusing on the problem of quantum–classical correspondence. In this paper, the dynamics of this system are tested by two related yet different indicators: the time autocorrelation function of the canonical position and the out-of-time correlator of position and momentum.

## 1. Introduction: The Classical Limit of Quantum Mechanics and the Correspondence Principle and Decoherence

Since the very beginning of quantum mechanics, the problem has been posed about the so-called *classical limit*, if for nothing else to give a meaning to the measurement process [1]. In parallel, the *correspondence principle* was introduced. This principle comprises two different concepts which are closely intertwined. They appear clearly stated by Dirac in [2]: says he that classical dynamics is employed in Bohr’s theory via “*[…] the assumption, called the Correspondence Principle, that the classical theory gives the right results in the limiting case when the action per cycle of the system is large compared to Planck’s constant h*” Additionally, a few lines after: “*In a recent paper, Heisenberg puts forward a new theory which suggests that it is not the equations of classical mechanics that are in any way at fault, but that the mathematical operations by which physical results are deduced from them that require modification”.* Therefore, correspondence can be interpreted in two ways: according to the latter sentence, classical canonical variables can be turned into quantum mechanical operators and classical Hamilton equation into Heisenberg’s; according to the former, once this correspondence has been established, quantum dynamics should yield classical results for large values of the action. Dirac concludes the quoted paragraph by adding: “*All the information supplied by the classical theory can thus be made use of in the new theory* (emphasis in the original)”.

In a series of papers with Ford in the last century we raised the doubt that not *all* the dynamical content of classical dynamics can be retrieved by quantum dynamics in the classical limit [3,4]. In parallel with this theoretical question is the more practical attitude of considering the specific details of such correspondence: in other words, to determine which regimes and with what accuracy agreement of the two results are to be expected, and finally, to ask whether this formal correspondence (formal, because it can be stated at a fully abstract level) is capable of describing physical phenomena predicted by classical dynamics.

This situation can also be described as follows: while the correspondence principle affirms that the mathematical operations of quantum mechanics yield results that, for increasing semiclassical parameters, approach the results of *corresponding* mathematical operations of Hamiltonian dynamics, this approach is far from trivial, as remarked by Berry [5,6]. In fact, it involves two non-commuting procedures: a semiclassical limit that can take various forms, such as the Planck constant *h* tending to zero (that is clearly an oxymoron, for a constant does not vary) or, as demonstrated in this paper, studying particles of larger and larger mass, and a long-time limit, in the time-span of both evolutions. Then, formal correspondence must be reformulated as a question of the scaling relations between time, a semiclassical parameter, and the (properly formulated) difference between classical and quantum results [7].

To be sure, a particular feature of this relation was immediately realized in the early years of the study of *quantum chaos*: when considering the time evolution of a classically chaotic system, the combination of the Heisenberg uncertainty principle and the Lyapunov instability of classical motion restricts agreement of the two dynamics to times logarithmically short in the semiclassical parameter (the Berman–Zaslavky time, sometimes also named after Ehrenfest) [8]. Or, reversing the terms in the equation, to achieve coincidence over *linearly* increasing spans of time, an *unphysical, exponential* increase in the semiclassical parameter is required. Albeit previously realized by Chirikov, Percival, Berry, Kolovsky, and others, perhaps the most catchy representation of this fact has been exhibited by Zurek and Paz by considering the chaotic motion of Hyperion [9,10,11], showing that the Berman–Zaslavsky time for this celestial object is much too small to agree with observations. The resolution of this paradox has been proposed, invoking the phenomenon of decoherence [12,13,14], which, in essence, limits the resolution of phase-space by blurring quantum (and classical)evolution via interaction with an environment. An alternative view that is not examined here was presented in [15]. The historical origins of the decoherence approach are discussed in [16,17]: particularly relevant to my view are [18,19,20,21,22,23,24,25,26,27,28,29,30,31].

Since the notion of decoherence invokes the interaction of a system with its environment, many models have been proposed for this aim. Most of them turn the evolution of the density matrix into a master equation in Lindblad form [32]. In this paper, the goal of representing this interaction exactly via the model of the multi-particle Arnol’d cat is pursued [16]. This model has a fundamental difference with respect to most models: it is fully unitary, and as such, it is immune to dissipation. Moreover, it avoids recourse to any approximation, or the introduction of infinite baths of oscillators, or even random dephasing of the quantum evolution: these are sound procedures, but they input into the system an infinite amount of information which hinders the possible organization of information coming from the exact Hilbert space dynamics. To appreciate this otherwise obscure remark, the reader is referred to the more complete discussion in [16,17]. Furthermore, the proposed system gains more relevance because it permits the deriving of *scaling laws* in terms of the physical parameters (that shall be described momentarily) that cannot be easily disentangled by the usual approaches mentioned above.

Now, let me introduce briefly the content of this paper. The Hamiltonian of the multi-particle Arnol’d cat, briefly described in Section 2, is that of a free particle on a ring subject to periodic impulses (*kicks* in jargon) that change its momentum instantaneously [3] in such a way that the Floquet map of the classical motion is the Arnol’d cat mapping [33]. Moreover, this *large* particle is coupled via elastic scattering to a number of smaller particles, also rotating in the ring [16]. This coupling and the full evolution can be computed to any desired accuracy by a precise numerical technique. Then, we assume that we are allowed to measure the dynamical variables of the large particle only, which constitutes the *system*, while the remaining particles are the *environment*, and yet we treat the two as a whole quantum system. An early example of this procedure is Kolovsky’s treatment of two coupled standard maps on the torus [34].

It is then important to examine the dynamics of this system in light of the commonly accepted indicators of dynamical complexity. In my view, the most important concept in this category is quantum dynamical entropy [35,36,37], which *corresponds* to the Kolmogorov Sinai classical entropy [38,39]. Its limited appearance in the literature is most probably due to the fact that it is extremely difficult to compute, numerically and even in simple models [36,37,40]. I studied this entropy in [16,40] and in a previous paper in this journal [17]. Furthermore, in [41], I focused my attention on the exact Von Neumann entropy (not the linear version of it) of the reduced density matrix, obtained by tracing the degrees of freedom of the small particles. In many investigations (see the recent [42] and references therein), the initial growth of this quantity has been related to classical Lyapunov exponents. While permitting us to “observe” visually the decohering effect on *Schrödinger cat*s wave function [43], the study of the multi-particle Arnol’d cat system raises some caveats on the above conclusions because it shows that careful scaling of the physical parameters is required to achieve the desired result. In addition to dynamical entropy, various attempts have been made to introduce a quantum notion of Lyapunov exponents [44,45,46,47]. They also reveal the limitations that quantum dynamics pose to the development of true dynamical instability.

The aim of the present work is to continue the analysis of the multi-particle Arnol’d cat by investigating two quantities that come in the form of the trace of products of dynamical operators corresponding to classical canonical variables. Again, we consider the position and momentum operators *Q* and *P* corresponding to the large particle. The first quantity, considered in Section 3, is the time autocorrelation function (*ACF* for short) of the position *Q*, $C_{Q}\left(t\right)=\mathrm{Trace}Q\left(t\right)\rho Q$, where ρ is an invariant density matrix and Q(t) is the time evolved operator $Q\left(t\right)=U^{-t}\;Q\;U^{t}$. Here, $U^{t},\;t\in R$ is the group of quantum evolution operators.

This quantity, when *Q* is replaced by a generical dynamical variable subject to certain conditions, has been studied long ago [48,49] in the case of the kicked rotor, showing correspondence only for times logarithmically short in the semiclassical parameter, and power-law decay with special features afterwards. A different result was found for the evolution generated by symmetric matrices (Hamiltonians) in the GUE ensemble [50]. It was shown that the average of ACFs over the statistical ensemble of matrices tends to be a constant, bottom value that depends on the size of the matrix N (that is, the dimension of the Hilbert space). This value vanishes when N increases, and so does the dispersion of the autocorrelation functions generated by individual Hamiltonians. Furthermore, the initial decay is extremely rapid, and it takes place on a time span that also vanishes when increasing N. In short, *random matrix dynamics approach a pure Bernoulli-type behavior* [50].

In this paper, I investigate the position ACF in four dynamical situations: when the large particle is free (i.e., not subject to periodic kicks or too small particle scattering), when kicks are switched on, but the scattering cross section is null, when this situation is reversed, and finally, when both kicks and scattering are present. Comparison with the random matrix result just quoted and with the classical theory will be presented.

Summarizing the obtained results, we will find that the motion of the quantum free particle corresponds to a linear transformation of the torus in such a way that, at any finite precision, agreement between classical and quantum results extends for a time range that scales linear to the mass of the particle: the correspondence principle is physically vindicated here. Next, the effect of the scattering perturbation will be shown to enhance the decay of the ACF in the free particle case, but more importantly, to quench the periodic resurgences of the quantum Arnol’d cat in such a way to reproduce the random matrix behavior quoted above.

In the second part of the paper, focus is on the out-of-time correlator (commonly abbreviated as *OTOC*) of position and momentum, which is also defined in terms of traces of product of operators: $O\left(t\right)=\mathrm{Trace}\left[Q\left(t\right),P\right]\;\rho \;{\left[Q\left(t\right),P\right]}^{†}$, where [·,·] is the quantum commutator. The short time behavior of O(t) has been put in relation to classical Lyapunov exponents [51,52,53,54]. I perform its analysis in the same dynamical situations described above. The results will show that the OTOC is constant in time for the quantum free particle, different to other dynamical indicators. In this case, scattering will be shown to contribute an increase to the OTOC that is proportional to the intensity of the perturbation: this result is instrumental to provide a heuristic explanation of the experimental data of the OTOC for the full dynamical system, both before and after the Berman–Zaslavsky time scale.

In the Conclusion, I briefly comment on the significance of the obtained results.

## 2. Review of the Multi-Particle Arnol’d Cat

The multi-particle Arnol’d cat is a generalization of the classical Arnol’d cat when viewed as a particle freely rotating in a ring and subject to a periodic impulsive force. It is obtained by adding a number of additional particles in the ring, elastically scattering with the former. This construction accomplishes the main goal of rendering the Joos–Zeh approach to decoherence [20] in a fully unitary and controllable way. In fact, in doing so, the introduction of an infinite amount of information provided by infinite baths of oscillator is avoided, as well as dissipative effects present in other models: see [16] and a previous paper in this journal [17] for further comments on this topic.

The Hamiltonian of a particle of mass *M* that we consider is
(1)$$H_{cat}\left(P,Q,t\right)=\frac{P^{2}}{2M}-\eta \frac{Q^{2}}{2}\sum_{j=-\infty }^{\infty }\delta \left(t/T-j\right),$$
where variables *Q* and *P* take value in the unit torus, and *T* is the period between kicks of amplitude η. If parameters are chosen as we will do momentarily, the classical evolution generated by this Hamiltonian over time *T* is the Arnol’d cat map. Notice that classical chaos in generated by the *inverted* harmonic potential in Equation (Equation 1) for η>0. On the contrary, when η<0, the Hamiltonian generates an elliptic map, whose behavior is more akin (i.e., stable) to the free motion η=0 discussed later in this paper.

Quantization of this and of similar systems have been studied over the last century [3,55,56,57,58] and can be effected by a combination of kinematics and dynamics on the space of square summable periodic functions in both position and momentum [59,60]. The particular Hamiltonian Equation (Equation 1) has been quantized in this way in [3], successive to the original treatment by semiclassical means in [58]. The canonical formalism also permits to write a convenient finite-dimensional Wigner function [3,58]. This approach has been fruitfully applied (and some formulae rediscovered) in many successive works.

In short, the Hilbert space of the one-particle system Equation (Equation 1) is unitarily equivalent to $C^{N}$, where the dimension N is proportional to the mass *M* of the particle, which is free to be increased, to enter the semi-classical region, while the Planck constant *h* is *kept constant*, as it should be as follows:(2)$$\frac{ML^{2}}{T}\;=\;Nh.$$
In the above, L=1 is the periodicity of the torus in the *Q* direction and *T* is the time interval between kicks, also set to unity: see [3,16] for more detail. To regain the classical Arnol’d cat, the dimensional paramenter η in Equation (Equation 1) is also given the value one. It is nonetheless important that physical quantities carry correct dimensions. This is because, in Equation (Equation 2), the mass *M* of the particle is quantized.

Evolution in this space is effected by a finite, unitary matrix *U*. Following [3], the matrix components of *U* in the position representation are as follows. The free evolution operator $U^{free}$ has matrix elements:(3)$$U_{kl}^{free}=\frac{1}{\sqrt{N}}e^{-(\pi il^{2}/N)}e^{2\pi ikl/N},$$
while the kick operator *K* is as follows:(4)$$K_{kl}=\frac{1}{\sqrt{N}}e^{i\pi l^{2}/N}\delta_{k,l},$$
where the indices k,l range from 0 to N−1, $\delta_{k,l}$ is the Kronecker delta, and finally, $U\;=\;KU^{free}$.

Perhaps the main advantage of the canonical formalism over the semiclassical is that it allows a straightforward extension of the model by adding a number *I* of freely rotating, smaller particles of mass *m* onto the ring, scattering with the former via a point interaction V(·) that is united when its argument is null and zero otherwise. The Hamiltonian of the full system [16] is, therefore,
(5)$$H=H_{cat}\left(Q,P,t\right)+\sum_{i=1}^{I}\frac{p_{i}^{2}}{2m}+\kappa \sum_{i=1}^{I}V\left(q_{i}-Q\right).$$
The constant κ is a sort of cross section for the scattering of the large particle and the smaller ones.

Quantum kinematics for the small particles is formally achieved along the same lines as for large particles: In addition to the L=1 periodicity in the variables $q_{i}$, we also impose periodicity $\frac{mL}{T}$ in the momenta $p_{i}$. We obtain that the mass of the small particles is also quantized:(6)$$\frac{mL^{2}}{T}=\nu h,$$
where ν is an integer. For convenience, we choose both ν and N to be powers of two. Furthermore, being the allowed position of large and small particles restricted to a lattice, we stipulate that each coarser lattice (for the small particles) is a subset of that of the large: the allowed positions are as follows:(7)$$\left(Q,q_{1},\ldots ,q_{I}\right)=\left(\frac{j_{0}}{N},\frac{j_{1}}{\nu }+\frac{s_{1}}{N},\ldots ,\frac{j_{I}}{\nu }+\frac{s_{I}}{N}\right),$$
in which $j_{0}$ ranges from 0 to N−1, and the index i=0 is related to the particle of mass *M*; $j_{i}$ ranges from 0 to ν−1 and labels the position of a small particle, and *i* ranges from 1 to *I*; finally, the integer $s_{i}$ provides the shift of the origin of the *i*-th lattice with respect to the zeroth. Terms at the right-hand side of Equation (Equation 7) are meant to be modulus one.

The above expression also provides a basis for the Hilbert space of the system, in the position representation, as a set of delta functions at the positions in Equation (Equation 7). This space is, therefore, isometric to a power of C:(8)$$H=H_{cat}⊗H_{small}^{I}=C^{N\nu^{I}}.$$

According to the choice that *V* be a local interaction, in the position representation its matrix elements are
(9)$$V_{j_{0},j_{1},\ldots ,j_{I};j_{0}^{\prime },j_{1}^{\prime },\ldots ,j_{I}^{\prime }}=\prod_{i=0}^{I}\delta_{j_{i},j_{i}^{\prime }}\sum_{i=1}^{I}\delta_{j_{0},\frac{N}{\nu }j_{i}+s_{i}}.$$

As explained in [16], the unitary evolution in this $N\times \nu^{I}$ dimensional space is effected by a Trotter–Suzuki product technique. For the computations in this paper, we employ the second-order approximation version. These computations can be carried on to any desired degree of accuracy.

To define observable quantities, following standard usage we replace *Q* and *P* with periodicized quantum operators on the two-torus, $Q^{per}=\sin\left(2\pi Q\right)$ and $P^{per}=\sin\left(2\pi P\right)$. These operators are diagonal in the position and momentum representation, respectively. It is important to remark that they act on the coordinates of the *large particle* only, so that the corresponding diagonal elements are as follows:(10)$$Q_{jj}^{per}=\sin\frac{2\pi j}{N},\;\;P_{kk}^{per}=-\sin\frac{2\pi k}{N},$$
where *j* and *k*, ranging from 0 to N−1, label the position and momentum of the large particle in the corresponding representations. For easiness of notation, in the following we drop the periodicity superscript. Unitary transformation between the two representations is performed by a discrete Fourier transform conveniently coded by a fast Fourier numerical scheme (which must obviously comprise the tensor product Hilbert space in Equation (Equation 8)). Computations are faster when choosing N to be a power of two, as is obvious.

## 3. The Autocorrelation Function of the Position

As described in the introduction, to investigate the dynamics of the multi-particle Arnol’d cat, we first consider the time autocorrelation function of the position, defined as follows:(11)$$C_{Q}\left(t\right)=\langle Q\left(t\right)\;\rho \;Q\rangle ,$$
where *Q* is the periodicized position operator acting only on the large particle variable, Equation (Equation 10). In terms of the evolution operator, assuming ρ to be the identity divided by the dimension of the Hilbert space, so to correspond to the Lebesgue measure on the torus that is classically invariant, this becomes the following:(12)$$C_{Q}\left(t\right)=\langle U^{-t}\;Q\;U^{t}Q\rangle ,$$
where the following 〈·〉 indicates the trace divided by the Hilbert space dimension.

### 3.1. ACF for the Free Particle

We first study the function $C_{Q}\left(t\right)$ (we drop the subscript *Q* for conciseness) when the large particle is alone (I=0) and *not* subject to the Arnol’d kick (η=0). The classical dynamics on the two-torus are given by the invertible map:(13)$$\Phi^{t}\left(Q,P\right)=\left(Q+Pt,P\right)\;\mathrm{mod}1.$$
The correlation function of the periodic position is given by the integral
(14)$$C_{cl}\left(t\right)=\int \;\int \sin\left(2\pi Q\left(t\right)\right)\sin\left(2\pi Q\right)\;dQ\;dP,$$
which can be easily integrated to show that the classical ACF is given by half the function sinus cardinal:(15)$$C_{cl}\left(t\right)=\frac{\sin(2\pi t)}{4\pi t}=\frac{1}{2}\mathrm{sinc}\left(2\pi t\right).$$
As it is apparent, this function features periodic oscillations mitigated by slow, power-law decay. In the following figures we find it more illustrative to plot the value of ACF multiplied by two, since the average value of $\sin^{2}\left(2\pi Q\right)$ is one half. In so doing the ACF at time zero takes the value one.

Next, we compute the quantum ACF for finite N and κ=0 and compare it with the classical value Equation (Equation 15). Firstly, it is to be remarked that the quantum ACF cannot feature the same decay as the classical, since it is a periodic function of time. In fact, let us consider the free evolution over an integer span of time *t*. According to the choice of dimensional constants presented in Section 2, which leads to Equation (Equation 13), the motion is simply given by Q(t)=P(0)t+Q(0), P(t)=P(0). As mentioned before, the evolution of the Wigner function at integer times is governed by the classical map on a N×N lattice because of a commutative diagram permitting to retrieve one from the other (see, e.g., Equation (37) in [3]), with the sole difference of a phase factor. As a consequence, when *t* is a multiple of N, the quantum motion retrieves the initial condition (modulus a phase that can affect the sign of the ACF) and forcefully cannot follow the classical decay in Equation (Equation 15), as seen in Figure 1.

Nonetheless, quantum motion does approximate the classical better and for longer time spans as the mass *M* and, therefore, N grows. While this analysis can be performed analytically to a certain extent, we prefer to carry it out numerically, also to better compare its results with those of the next sections, when we will compute the ACF in the case of the multi-particle Arnol’d cat.

Therefore, we consider a time span shorter than the periodicity t=N=M/h. In Figure 2, we plot $C_{cl}\left(t\right)$, the quantum C(t), and the difference $\Delta C\left(t\right)=C\left(t\right)-C_{cl}\left(t\right)$ for increasing values of *M* and two time windows: the first (left panel) at the beginning of the evolution, and the second between t=16 and t=20. We observe that, as expected, ΔC(t) diminishes as *M* grows, while, albeit oscillating, it is roughly constant or at most slowly increasing over time. To quantify the validity of this observation in Figure 3, we compute ΣΔC(t), which is the integral of |ΔC(s)| from s=0 to s=t, when $N=2^{6},2^{7},2^{8},2^{9}$. These values are indicative of an average linear increase in ΣΔC(t) with time. Moreover, when plotting the rescaled quantity N×ΣΔC(t) as in the upper part of Figure 3, we find that the different curves collapse on a single law of the kind h(t)=At. This leads to the conclusion that
(16)$$\Delta C\left(t\right)∼\frac{Ah}{M}+o\left(1\right),\;\mathrm{for}\;t\ll \frac{M}{h},$$
where *A* is a constant independent of M/h.

Suppose now that we fix an upper bound δ to ΔC(t). The mass *M* required to satisfy this bound must grow linearly in $\frac{1}{\delta h}$, and the time-span of agreement grows linearly with $\frac{M}{h}$. This is the well-known conclusion held for integrable or almost integrable systems: the classical limit of quantum mechanics can be achieved with a physically sound increase in the semiclassical parameter.

### 3.2. ACF for Free Quantum Motion in the Presence of Scattering

We now introduce the effect of scattering on the motion of the free quantum particle, i.e., in the absence of the cat kick (η=0). In this case, the ACF is a quasi-periodic function of time, but due to the exponentially increasing dimension of the Hilbert space with the number of small particles, we expect periodicities or almost periodicities to manifest themselves far in the time evolution. Actually, the dimension count in Equation (Equation 8) is one of the original motivations behind the introduction of this system.

In Figure 4, we plot the correlation function C(t) in various dynamical situations, characterized by the same value of *M* and *m* and the same number of small particles (therefore, the same “quantum vs. classical” situation), but increasing coupling κ. Also plotted is the classical value $C_{cl}\left(t\right)$ (which can be thought of as the M=∞ case). The two panels in the figure show the correlation function at the beginning of the evolution and after ten cycles of unperturbed motion. Observe the different vertical scales in the two panels. Increased decay of the correlation function for increasing perturbation by the small particles is observed.

To better gauge this effect, we compute Σ|C(t)|, the integral of the absolute value of the correlation function up to time *t*
(17)$$\Sigma |C\left(t\right)|=\int_{0}^{t}\left|C\left(s\right)\right|\;ds.$$
In the case of the free particle of infinite mass (that is, in classical dynamics), this function is logarithmically divergent, a first year calculus exercise. Therefore, when plotted versus the logarithm of time, it features a linear graph. In Figure 5, the same law appears to characterize, in the investigated range, the free rotation of mass $M=2^{8}h$, already described in Figure 2. On the contrary, when coupling is effective, the correlation decay observed in Figure 4 is revealed by the slower increase of Σ|C(t)|, suggesting that, in this range, this function can be parameterized as follows: $S\left(t\right)=A_{\kappa }\left(1-B_{\kappa }t^{1-d_{\kappa }}\right)$, with $d_{\kappa }$, $A_{\kappa }$, and $B_{\kappa }$ as suitable constants. This functional form applied to the case when three particles of mass 2h collide with the large particle of mass $M=2^{8}h$ yields a good fit with $d_{\kappa }≃1.66$, when κ=64 (shown in the figure). Similar data sets in Figure 5 are fitted by $d_{\kappa }=1.15$ for κ=32 and $d_{\kappa }=1.92$ for κ=256. As a consequence, in this range, the ACF behaves as follows:(18)$$\left|C\left(t\right)\right|∼B_{\kappa }t^{-d_{\kappa }}.$$

A reviewer suggested to put this result in relation to the anomalous diffusion characterizing a large family of classical systems [61,62]. In the quantum domain, the spreading of wave-packets governed by power-laws with many exponents is the content of the so-called theory of quantum intermittency, see [63] and references therein, that has the flavor of a spectral theory, unrelated to a classical analogy. In the present context, a possible explanation of the faster decay of correlations can be obtained by a purely classical heuristic argument, as follows. For totally free motion, the *sinc* decay comes from the shear of layers of phase space at different values of *P*. Collision with the additional particles has the effect of changing the momentum of the large particle, moving its phase-space coordinates to a different layer with different velocity, a fact that enhances the divergence of nearby trajectories. A model for this behavior was studied in [64], in connection with flow in porous materials.

To summarize, the effect of the environment is to increase the exponent of the power-law decay of the quantum ACF with respect to the unperturbed classical, in a time range that scales as $t\ll \frac{M}{h}$, as in the previous subsection. Presumably, the same behavior is also to be found in the classical dynamics of the many-particle system, which is not studied in this paper.

### 3.3. ACF for the Multi-Particle Arnol’d Cat

Finally, let us include the cat kick (η=1), so that the full dynamical system is considered. As mentioned before, the quantum Arnol’d cat is characterized by the same periodicities of the classical cat on a N×N lattice because of the commutative diagram relating it to the quantum evolution of the Wigner function (see, e.g., Equation (37) in [3]). Yet, it is seen that, except for these periodicities, the autocorrelation function of the quantum cat is null at integer times that are not multiple of the period; indeed, the situation is as shown in Figure 6. On the $2^{4}\times 2^{4}$ lattice, the classical cat is periodic of period τ=12. The autocorrelation function is symmetric around zero and null at time one. The behavior for t∈[−1,1] is periodically repeated for t∈[−1+lτ,1+lτ] and for the *l* integer.

Simple computations in modular arithmetic show that, when $N=2^{j}$, like in the examples in this paper, the period $\tau_{j}$ is given by the following formula:(19)$$\tau_{j}=3\cdot 2^{j-2}\Rightarrow \tau \left(M\right)=\frac{3}{4}\frac{M}{h};$$
that is, the period grows linearly to the mass *M*. See [65,66] for the general case and [67] for the first picture of this phenomenon. The quantum Arnol’d cat somehow achieves what happens for the evolution generated by random matrices, as discussed in the Introduction, albeit it reveals its algorithmic simplicity by these recurrences.

Coupling with additional particles has the effect of preventing such recurrences, as shown in Figure 6. Particularly instructive is the curve obtained for I=3 and κ=1. In fact, we see that, while in the zero period the curve quite approximately reproduces the κ=0 case (i.e., the original quantum cat), in the first period, the periodic peak reaches an amplitude of approximately 0.8, further diminishing to roughly 0.4 in the second. This quenching effect is greatly amplified as the intensity κ is augmented (curves with κ=2,4,8,16).

The multi-particle quantum cat ACF is, therefore, a dynamical realization of the behavior of the ensemble averaged GUE Hamiltonians.

## 4. The Out-of-Order Commutator of Position and Momentum

The correspondence principle has been employed to unveil the dynamical importance of the out-of-order commutator between the two self-adjoint operators *Q* and *P*. In fact, let O(t) be defined as follows:(20)$$O\left(t\right)=\langle \left[Q\left(t\right),P\right]\;\rho \;{\left[Q\left(t\right),P\right]}^{†}\rangle ,$$
where 〈·〉 is the trace operation in the Hilbert space of a system, ρ is a density matrix, and $Q\left(t\right)=U{\left(t\right)}^{†}QU\left(t\right)$, U(t) is the unitary evolution group. Then, if *Q* and *P* take on the classical interpretation as canonical position and momentum coordinates, replacing the quantum commutator [Q(t),P]=Q(t)P−PQ(t) by the classical Poisson bracket $\left\{Q\left(t\right),P\right\}=\frac{\partial Q(t)}{\partial Q}\frac{\partial P}{\partial P}-\frac{\partial Q(t)}{\partial P}\frac{\partial Q(t)}{\partial P}$, immediately yields
(21)$$\left\{Q\left(t\right),P\right\}=\frac{\partial Q(t)}{\partial Q},$$
which calls into play the local maximal Lyapunov exponent at *Q*: $\left\{Q\left(t\right),P\right\}∼e^{\lambda (Q)t}$. One should, therefore, expect that the quantum O(t) should behave accordingly, as the exponential of time multiplied by twice the classical Lyapunov exponent.

From this heuristic remark, one immediately observes that correspondence requires ρ to be a pure state $|\psi_{Q}\rangle \langle \psi_{Q}|$ of minimal uncertainty at position *Q*. The obvious dependence on *Q* calls into play the well-known classical large deviation theory of Lyapunov exponents. A detailed study of the large deviation function for quantum Lyapunov exponents has been recently published in this same journal [54]. It is also to be remarked that different actions, such as taking the logarithm of O(t), tracing the logarithm of square commutator, or averaging the latter over different $|\psi_{Q}\rangle \langle \psi_{Q}|$ as in [53], give different results, as explained in [51]. In this work, we do not investigate such alternative procedures but stick to the definition in Equation (Equation 20).

The validity of the short time relation to Lyapunov exponents has been confirmed in a series of works (see, e.g., [51,52,53,54] and references therein), all exhibiting breakdown of correspondence at the Berman–Zaslavsky time scale. In line with the discussion in the introduction, it is natural to ask what decoherence can add to the picture. We can start from a formal argument.

We assume that the density matrix ρ is the identity divided by the Hilbert space dimension since this yields the natural analogue of the invariant Lebesgue measure on the classical two-torus. Then, O(t) in Equation (Equation 20) is equal to the difference of two real quantities $O\left(t\right)=O_{+}\left(t\right)-O_{-}\left(t\right)$, given by the following:(22)$$O_{+}\left(t\right)=2\;\langle P^{2}Q^{2}\left(t\right)\rangle ,\;\;O_{-}\left(t\right)=2\;\langle PQ\left(t\right)PQ\left(t\right)\rangle ,$$
more specifically
(23)$$O_{+}\left(t\right)=2\;\langle P^{2}\circ U^{-t}\circ Q^{2}\circ U^{t}\rangle ,\;\;O_{-}\left(t\right)=2\;\langle P\circ U^{-t}\circ Q\circ U^{t}\circ P\circ U^{-t}\circ Q\circ U^{t}\rangle .$$
The composition symbol ∘ in the above is superfluous, but it enhances the readability of these and successive formulae. Let us compare these relations with the decoherence matrix $D_{\theta ,\eta }$ that is central in the notion of dynamical entropy, employed in the previous paper in this journal [17]. From Sect. III, we see that $D_{\theta ,\eta }$ is the trace of the sequence of operator products
(24)$$\Pi_{\theta_{0}}\circ U^{-1}\Pi_{\theta_{1}}\circ \cdots \circ U^{-1}\Pi_{\theta_{n-1}}\circ \Pi_{\sigma_{n-1}}U\circ \cdots \circ \Pi_{\sigma_{1}}U\circ \Pi_{\sigma_{0}},$$
where $\Pi_{i}$ have been chosen as projection operators for *Q* in a partition of the unit interval by sub-intervals of equal size, and θ and η are symbolic words. We see that, in this approach, the symbolic history of the quantum motion is considered at successive time intervals while in Equation (Equation 23), only the initial and final states of the motion contribute.

Moreover, passing from form to substance, the results of [16,17] suggest, or at least leave the possibility open, that the interaction of a system with the environment could enable the former to *produce information* at the classical rate beyond the Berman–Zaslavsky time, and more importantly, for a range scaling linearly with the number of small particles representing the environment in the model under study. It is then natural to wonder whether the OTOC could also provide a tool to investigate this phenomenon.

### 4.1. OTOC for the Free Particle

It is instructive to notice that one can explicitly compute the OTOC in certain cases. For a single free particle, both $O_{+}\left(t\right)$ and $O_{-}\left(t\right)$ take a constant value:(25)$$O_{+}\left(t\right)=1/2,\;\;O_{-}\left(t\right)=\frac{1}{2}\cos\left(2\pi /N\right),$$
as it easily verified by direct computation. Therefore, O(t) is constant in this case, different to the dynamical complexity of the classical system, which increases logarithmically, as well as the quantum Alicky Fannes entropy [40] and the Von Neumann entropy [41]. This fact seems to already indicate a difference between the former indicators and the OTOC.

### 4.2. OTOC in the Presence of Scattering of the Large Particle by Smaller Ones

Let us now study the effect of scattering by small particles on the OTOC of the large particle variables, investigated via the autocorrelation function in Section 3.2.

In Figure 7, we plot O(t) (left panel) and logO(t) (right panel) versus time for exponentially increasing values of the coupling κ over a large range of values from κ=8 to κ=256 when $M=2^{8}h$, m=2h, and I=3. We observe that O(t) tends to be a limiting value O(∞) that depends on κ. For the largest coupling, wide oscillations of O(t) are observed, related to a strong perturbation of the motion of the large particle.

Let ΔO(t) be the increase of the logarithm of the OTOC from its initial value, which corresponds to the constant value of the OTOC for the free particle: logO(0)=log[1−cos(2π/N)]−log(2), approximately equal to:(26)$$\log O\left(0\right)≃2\log\frac{\pi }{N}=-2\log\frac{M}{h\pi },$$
as it follows from Equation (Equation 2). Therefore, we define
(27)$$\Delta O\left(t\right)\;=\;\log O\left(t\right)+2\log\frac{M}{h\pi }$$
and we use the asymptotic value ΔO(∞) to quantify the effect of the perturbation, which is seen to grow approximately proportional to κ: the fitting line in Figure 8 has slope a≃1.18:(28)$$\frac{O(\infty )}{O(0)}∼\kappa^{a}.$$
This information will be used later in Section 4.4 to explain the behavior of the OTOC for the full system: scattering yields a contribution proportional to κ to the OTOC, which is achieved within a short time span. Observe that the time unit is of the same order as the average time that a free particle needs to make a full turn of the ring.

### 4.3. OTOC for the Quantum Arnol’d Cat

Let us now turn to the one-particle quantum cat. The short time behavior of O(t) features an exponential increase with rate twice the classical Lyapunov exponent, as well established in many previous works (see, e.g., [51] and references therein). We consider in Figure 10 the natural logarithm of O(t), which at time zero takes the value in Equation (Equation 26). Albeit simply derived, this equation once more underlines the crucial feature of quantum mechanics brought about by the Heisenberg principle: the finest resolution of dynamical variables is dictated by N, which, in turn, *physically* corresponds to the mass of a quantum particle. In fact, observe in Equation (Equation 7) that the wave function of a quantum particle of mass *M* on the unit torus effectively lives on a regular lattice of spacing 1/N.

In the context of OTOC, the value in Equation (Equation 26) corresponds to the minimal uncertainty in the *Q* variable in Equation (Equation 21) at time zero, which increases exponentially immediately after. Even if repetitive of what has since long been described in the early works on quantum chaos, and verified in many successive papers, let us first study the behavior of this quantity when increasing exponentially the mass of the large particle in the absence of scattering (i.e., when κ=0). To do this, in Figure 9 we plot versus time the quantity ΔO(t) defined in Equation (Equation 26). It is observed that the curves initially coincide to high precision and the rate of exponential increase is twice the Lyapunov exponent. This situation lasts for larger spans of time the larger the value of the mass of the particle. Successively, they feature the typical interference behavior of the quantum cat. The explanation is quite simple: the scaling involved inEquation (Equation 27) the exponential growth of an initial difference in position stops when it has reached the (unit) size of the torus. This time is shorter the larger the initial difference. However, this is nothing new: this is the usual explanation of the Zaslavsky–Ehrenfest time scale.

In addition, the coincidence of the scaled curves is in favor of the *formal* validity of the classical limit: the larger the mass of the particle, the longer the correspondence of classical and quantum results holds. Nonetheless, recall the objection at this point: the range of this correspondence scales unphysically with the system parameters.

### 4.4. OTOC for the Multi-Particle Arnol’d Cat

The behavior of the OTOC seen so far is well known, see, e.g., [52], albeit we hope to have made it clear why our assessment of its physical significance is different from the common lore. But, let us now introduce coupling between the large particle and the smaller ones. We begin to expose our results by commenting on two figures with experimental results.

Firstly, in Figure 10, we plot the logarithm of O(t) for a particle of mass $M=2^{8}$, subject to the Arnol’d kick and scattered by *I* small particles of mass m=2. We consider various combinations of *I* and of the coupling constant κ that measures the intensity of the scattering. In the case of I=0, we recover the unperturbed quantum cat just studied.

Let us first consider the short time behavior. We observe that the effect of the perturbation in this logarithmic scale is more evident in the initial moments of evolution, while successively, the curves tend to realign themselves with the κ=0 case. The initial discrepancy is more noticeable the larger the scattering intensity, both in terms of the number of scattering particles *I* and of the coupling κ.

Next, in Figure 11, we plot the value of $O_{-}\left(t\right)$ at integer times in the same dynamical situations as in the previous figure. In all cases, the time zero value $O_{-}\left(0\right)$ is that of the single free particle, $O_{-}\left(0\right)=\frac{1}{2}\cos\left(2\pi /N\right)$.

The short time behavior can be defined here by positivity of $O_{-}\left(t\right)$ (and, therefore, t≤4 in the figure). In this region, the effect of scattering is to reduce the value of $O_{-}\left(t\right)$ with respect to the corresponding value with no scattering (I=0 or equivalently κ=0). Since the value of $O_{+}\left(t\right)$ is practically unaltered, this implies an increase of the value of O(t) with respect to the unperturbed cat value, observed by plotting the logarithm of the total quantity O(t) in the previous Figure 10.

We will consider the long time behavior of these quantities later on. Let us now try to frame theoretically the previous observations. In this regard, it is helpful to also display O(t) at non-integer times. Recall that impulsive perturbation is taking place at integer times, so that in between kicks, the dynamics of the large particle is only influenced by scattering by the smaller ones. In Figure 12, we repeat the analysis of Figure 10, now for a larger mass $M=2^{9}h$. For comparison, we again plot in the same figure the OTOC for the case I=3, κ=64 in the absence of the cat kick; that is, η=0 in Equation (Equation 1). Pay attention to the data for κ=64. We observe that the increase in the logarithm of OTOC taking place in between kicks, and therefore brought about by scattering alone, diminishes with time. When passing to a linear scale (not displayed), the increase in OTOC between kicks is approximately constant.

The analysis of Section 4.2 can now be used to propose a mechanism for these dynamics along the following lines. Firstly, scattering of the large particle by smaller ones produces a fixed perturbation in the unit time between kicks. Secondly, this perturbation adds to the divergence of trajectories; that is, it adds to the value of O(t) during each period between kicks. Thirdly, while this contribution is noticeable for short times, for larger times it is dwarfed by the exponential increase in O(t) entailed by the short-term chaotic behavior. This explains the behavior observed in Figure 10. In the conclusion, we will comment on the relation of this result with other theoretical approaches.

The above analysis can be made quantitative. We define the ratio of the quantity $O_{-}\left(t\right)$ for *I* and κ greater than zero with respect to the unperturbed quantum cat:(29)$$\rho_{\kappa }\left(t\right)=\frac{O_{-}\left(\kappa =0;t\right)}{O_{-}\left(\kappa ;t\right)}.$$
For vanishing values of the coupling κ, the ratio $\rho_{\kappa }$ tends to be one. Therefore, we find it convenient to consider the logarithm of $1-\rho_{\kappa }\left(t\right)$ as a function of time. This quantity is plotted in Figure 13, for $M=2^{8}h$, m=2h, I=3, and different values of κ and $\kappa =2^{j}$, with j=2,…,7.

This numerical analysis reveals that, for small values of time, $1-\rho_{\kappa }\left(t\right)$ grows exponentially with an exponent very close to one half and independent of κ over a large span of values:(30)$$\rho_{\kappa }\left(t\right)≃1-\exp\left\{-C+A\left(\kappa \right)+t/2\right\},$$
where *C* is a constant independent of *t* and κ. Furthermore, the term A(κ) scales approximately as κ to a power slightly larger than one, although the data do not allow us to determine a precise value (in the data, the exponent is roughly equal to 1.3). This provides a quantitative explanation to the behavior of the logarithm of O(t). In fact, let $B=\frac{1}{2}$, which is the constant value of $O_{+}\left(t\right)$. Let also δ(κ,t)=exp{−C+A(κ)+t/2}, and observe that, in the initial range of the data in Figure 12, it holds that δ(κ,t)≪1. Then, simple manipulations using the behavior of O(t) in the absence of scattering yield
(31)$$\log O(\kappa ;t)=\log O(0;t)+\log\{1+\delta \left(\kappa ,t\right)\;\left[B{\left(\frac{N}{\pi }\right)}^{2}e^{-2\lambda t}-1\right]\},$$
where λ is the classical Lyapunov exponent. At fixed time *t*, the κ dependence of δ(κ,t) implies that logO(κ;t) increases with κ. Next, consider the argument of the second logarithm at r.h.s. in Equation (Equation 31). It can be rewritten as follows:(32)$$\arg\log=1+\exp\left\{-C+A\left(\kappa \right)\right\}\left[B{\left(\frac{N}{\pi }\right)}^{2}e^{-(2\lambda -1/2)t}-e^{t/2}\right].$$
Now, let κ be constant. For small times the first term in the square bracket in Equation (Equation 32) is much larger than the second, so that when time grows the square bracket decreases and so does the difference logO(κ;t)−logO(0;t). This fully explains the short time behavior observed in Figure 12.

Turning finally to the behavior beyond the Berman–Zaslavsky time, one sees clearly in Figure 10 and Figure 11 that the effect of scattering is to quench the oscillations in the OTOC that are typical of the Arnol’d cat. This result can also be obtained by a modification of the one-particle quantum map [52].

## 5. Conclusions

The multi-particle Arnol’d cat is a simple yet complete system that I have introduced to analyze the nature and relevance of the decoherence approach to quantum chaos and to the correspondence principle. Based on this model, one can test various aspects of the dynamical behavior of quantum systems with a classical analogue. In particular, quantum dynamical entropies, as proposed by Alicky and Fannes [35,37], were investigated in [16,17,40] and the Von Neumann entropy in [41]. In this paper, I have studied the autocorrelation function of the position [48,49] and the OTOC, a quantity that has recently attracted considerable interest. Four dynamical situations have been investigated in both cases.

The first is that of a free particle. In this situation, the quantum ACF approximates the classical well in the sense that, given a prescribed degree of accuracy, the time span in which the two dynamics agree scales in a *physically acceptable* way in the system parameters. In turn, the OTOC is constant.

When scattering of the large particle with the smaller ones is considered, the ACF features enhanced power-law decay with an exponent larger than one, again for a physically acceptable range of times. Now, the OTOC initially grows and saturates at a value approximately proportional to the scattering intensity.

Next, I considered the standard quantum Arnol’d cat that has by now been thoroughly investigated in many works. Here, the ACF abruptly decays to zero, but with resurgences at periodic instants. In this case, the OTOC shows a previously well-known scaling behavior that once more highlights the relevance of Berman–Zaslavsy’s time scale. No new feature of this phenomenon is offered: it is presented to highlight the theoretical concepts in which I frame this result. To further clarify this *scaling* approach, which was first presented in [7], I refer the interested reader to an analogous family of phenomena occurring in planar classical billiards [68].

Finally, the full dynamics of the multi-particle Arnol’d cat have been considered. It was shown that by suppressing periodicity decoherence, it leads to an ACF behavior of the same kind as that of random matrices in the GUE ensemble [50]. In regards to the OTOC, decoherence affects the short time behavior of the OTOC and the parameter dependence of this phenomenon, heuristically and numerically estimated, confirms the following theoretical picture, which can be taken as a concise summary.

Quantum classical relation is best considered from a Wigner function perspective. In the family of systems that comprises the Arnol’d cat, this perspective yields classical motion on a square lattice, see [3,58]. Leveraging on this fact as a means to model quantum dynamics of the kicked rotor on the torus in the presence of external perturbation has been proposed [24,69], invoking random perturbations of the motion. The multiparticle Arnol’d cat described in this paper achieves this fact dynamically. Via its parameter dependence, it is possible to analyze physically the content of the correspondence principle, as described in Dirac’s quote in the Introduction without recourse to any approximation. In this respect, two indicators have been studied numerically herein, the autocorrelation function, which has a long history, and the more recent out-of-time correlator. A scaling picture of their behavior has numerically emerged, which must now be confirmed by a more rigorous investigation that should extend to an arbitrary, but finite, number of particles and their mass.

## Figures and Tables

**Figure 1 entropy-26-00572-f001:**
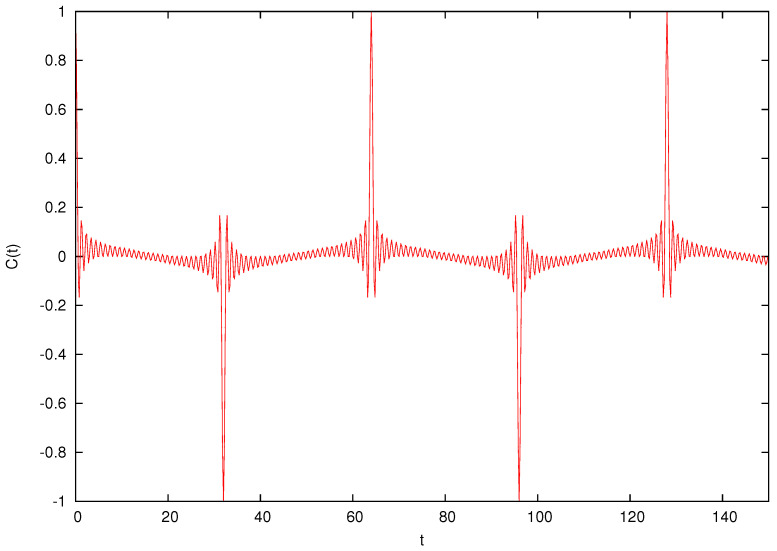
Position ACF C(t) for the quantum free particle when $M=2^{5}h$. The anti-periodicity at t=32 is evident.

**Figure 2 entropy-26-00572-f002:**
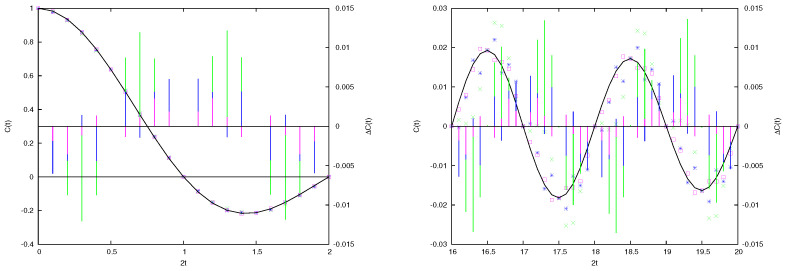
Autocorrelation function of position C(t) (left vertical scale) for the free particle of mass $M=2^{7}h$ (green crosses), $M=2^{8}h$ (blue stars), and $M=2^{9}h$ (red open squares). The continuous black line is the classical correlation function $C_{cl}\left(t\right)$ in Equation (Equation 15). Also plotted as vertical segments is the difference ΔC(t) between quantum and classical ACF (right vertical scale), same color coding of the symbols as before. In the left frame the time interval [0,1] is plotted, while in the right frame *t* belongs to [8,10]. Notice that as time increases, C(t) decays, but the difference ΔC(t) does not.

**Figure 3 entropy-26-00572-f003:**
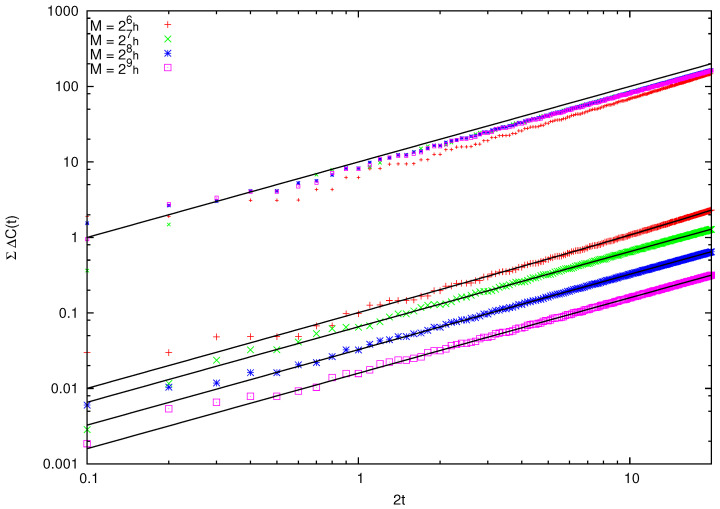
ACF of the large particle’s position in the free particle case. Integral of |ΔC(s)| from s=0 to s=t for $M=2^{6}h,\cdots ,2^{9}h$. The integral has not been normalized (i.e., shown is an arbitrary multiple of the integral, the same for all data sets). Fitting lines are a linear dependence. The top sets of data are the rescaled points M/h times ΣΔC(t), running parallel to the line h(t)=t/10. See text for discussion.

**Figure 4 entropy-26-00572-f004:**
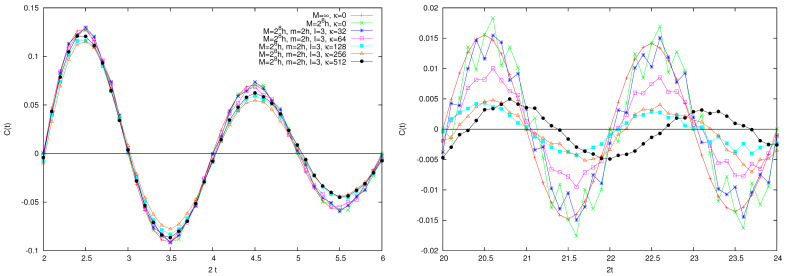
Autocorrelation function C(t) of the large particle’s position in the presence of scattering in various dynamical situations, as detailed in the legend. In the left frame the time the interval [1,3] is plotted, while in the right frame *t* belongs to the interval [10,12].

**Figure 5 entropy-26-00572-f005:**
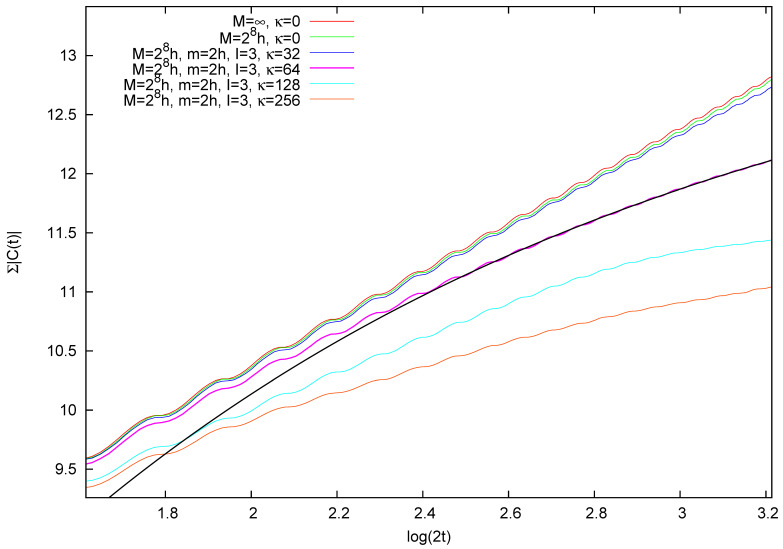
Autocorrelation function C(t) of the large particle’s position in the presence of scattering. Plotted versus time is the (non-normalized) integral of |C(s)| from s=0 to s=t in the dynamical situations listed in the legend. Fitting by the function S(t) described in the text has been performed in the interval log(2t)∈ [2.5:3.2]. The black curve refers to κ=64. See text for further discussion.

**Figure 6 entropy-26-00572-f006:**
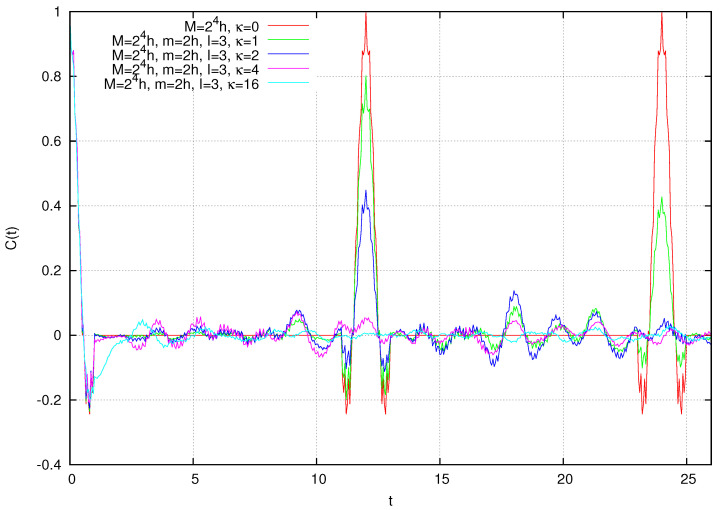
Correlation function C(t) for the multi-particle Arnol’d cat for various dynamical cases listed in the legend. See text for discussion.

**Figure 7 entropy-26-00572-f007:**
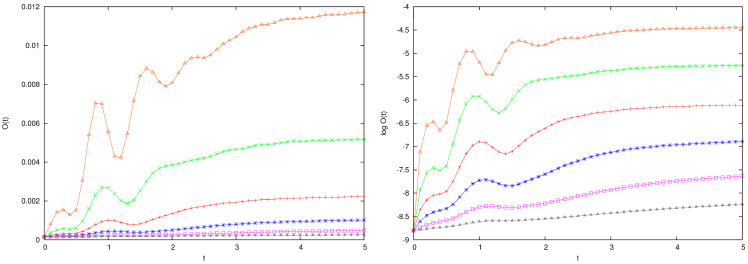
OTOC function O(t) for scattering of a large particle of mass $M=2^{8}h$ by I=3 particles of mass 2h versus time for κ=8,16,32,64,128,256, curves ordered from bottom to top. In the right frame, log(O(t)) shows the same situation.

**Figure 8 entropy-26-00572-f008:**
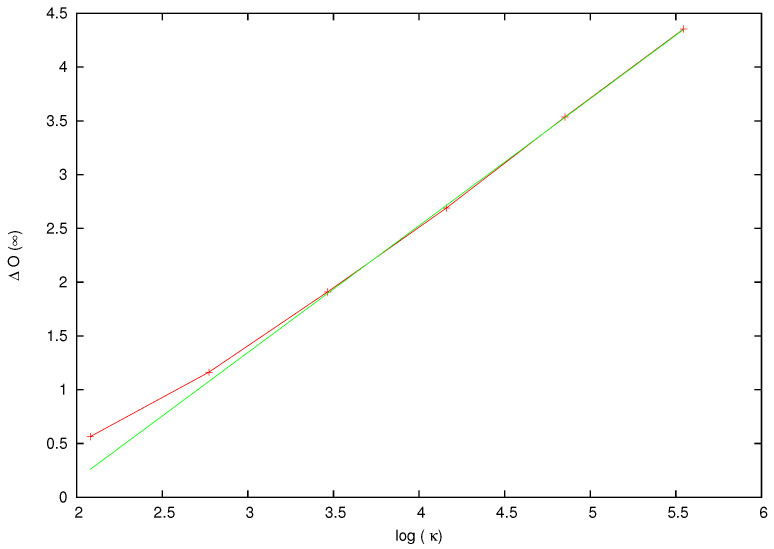
The asymptotic value ΔO(∞) defined in Equation (Equation 27) versus log(κ) for the data in Figure 7. The fitting line has slope a=1.18.

**Figure 9 entropy-26-00572-f009:**
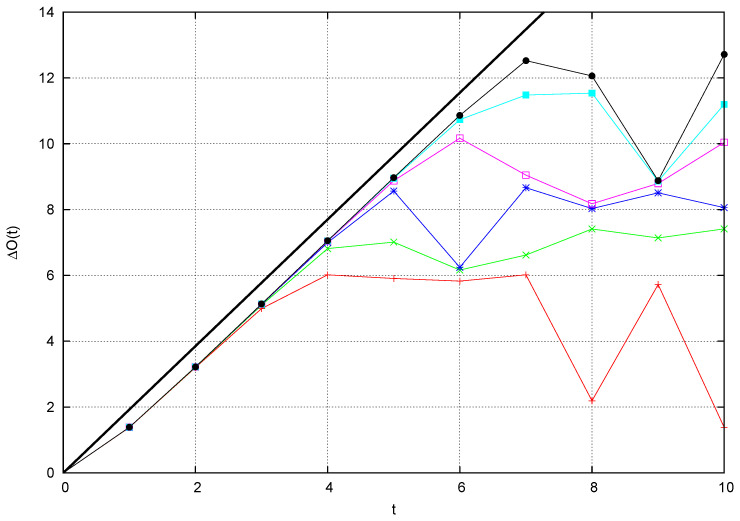
The scaled quantity ΔO(t) defined in Equation (Equation 27) versus time, for $M=2^{j}h$, j=6,…,11, curves increasingly ordered from bottom to top. The strait line has slope twice the Lyapunov exponent of the classical cat.

**Figure 10 entropy-26-00572-f010:**
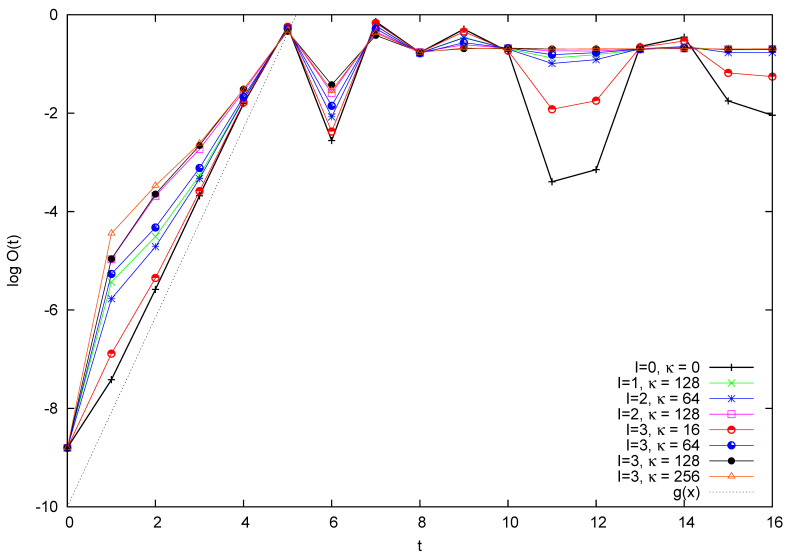
Logarithm of the OTOC function O(t) for $M=2^{8}h$, m=2h. The slope of the straight dotted line is twice the Lyapunov exponent of the classical cat. Different values of *I* and κ are investigated.

**Figure 11 entropy-26-00572-f011:**
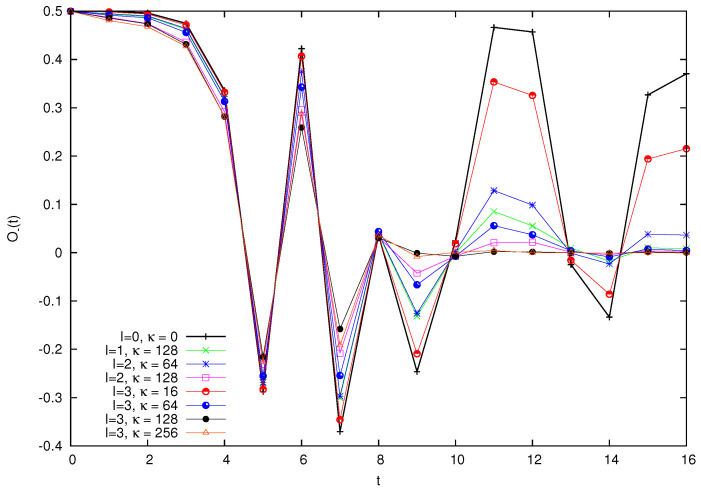
Quantity $O_{-}\left(t\right)$ for a particle of mass $M=2^{8}h$ subject to the Arnol’d kick and scattered by *I* small particles of mass m=2h. The scattering constant is κ. Values of *I* and κ are as in Figure 10.

**Figure 12 entropy-26-00572-f012:**
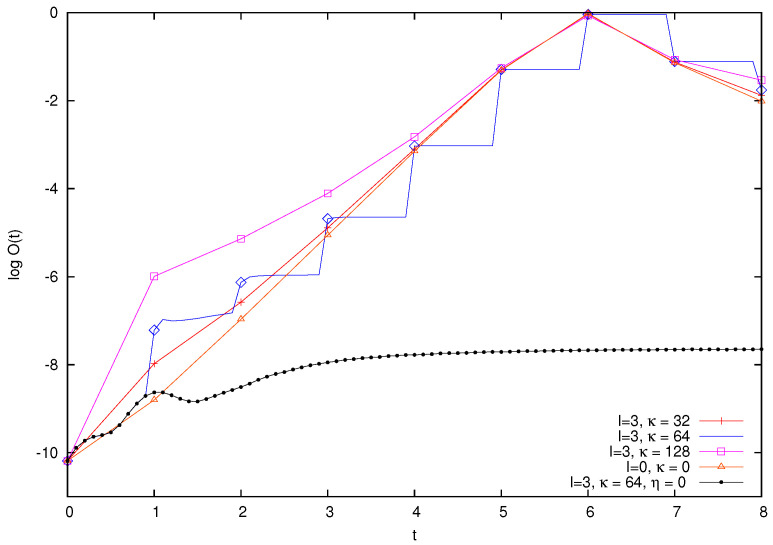
Logarithm of the OTOC function O(t) for $M=2^{9}h$, m=2h, and various values of *I* and κ. See text for explanation.

**Figure 13 entropy-26-00572-f013:**
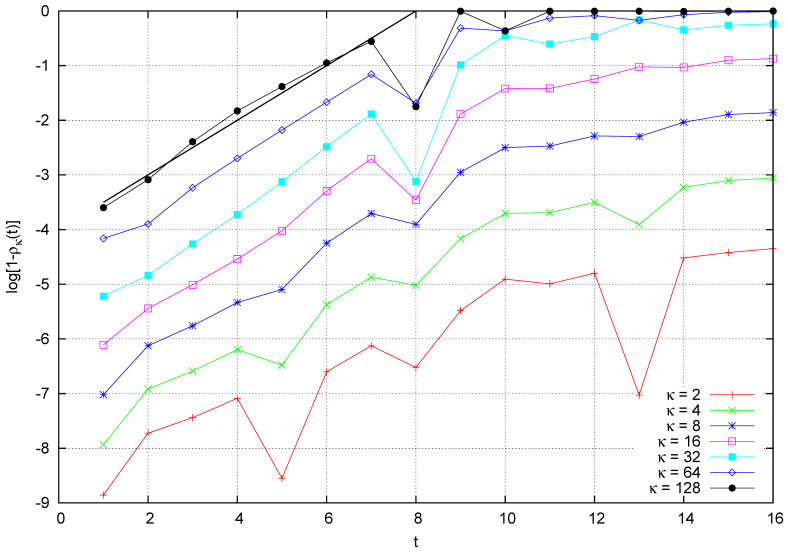
Logarithm of $1-\rho_{\kappa }\left(t\right)$ versus time, where $\rho_{\kappa }\left(t\right)$ is defined in Equation (Equation 29). Parameters are $M=2^{8}h$, m=2h, and I=3. The straight line has a slope on one half.

## Data Availability

The raw data supporting the conclusions of this article will be made available by the authors on request.

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
