# Peer review of "Behavior of Correlation Functions in the Dynamics of the Multiparticle Quantum Arnol’d Cat"

_entropy, 2024, doi:10.3390/e26070572_

Round 1

Reviewer 1 Report

Comments and Suggestions for Authors

Reviewer 2 Report

Comments and Suggestions for Authors

In this paper, the author extends his previous work on the dynamics of the multi-particle quantum cat map in the presence of mechanisms that allow for Hamiltonian 'internal decoherence' and the approach to the classical limit to study new metrics such as the autocorrelation function and OTOCs. 

The work is novel, solid, and the results should be of interest and use to the community. It is also generally presented well. I have no overall hesitation in recommending publication of this paper in Entropy.

However, some comments and suggestions:

(1) I would strongly recommend providing a short summary/preview/ some inkling of the results obtained, particularly in the context of their importance at the end of the introduction -- it currently is left hanging and carries more of a flavor of a report.

(2) In an otherwise excellent overview of the context for this work, I found it surprising that there was no discussion of or references to studies of Lyapunov exponents in quantum trajectories, nor to those that consider the relevance of Lyapunov exponents to entanglement generation (of relevance in this multi-particle case). I note that other groups have also studied quantum chaos and correspondence using Wigner functions in quantum cat maps.

A minor point: The references seem to jump from [15] to [30]

(3) Above eq 10 -- 'sinus cardinal' is a very unusual construction, and I may be missing something about what it means .. could the author clarify ?

(4) For Fig 2 it took me while to appreciate what the take-away point might be -- is it that the differences are bounded ? Could such take-aways be embedded in the captions ?

(5) In all the figures -- fig 4 onwards -- where there were various different dynamical systems and parameters considered it was not easy to understand what the range of 'physical' behaviors or systems were being explored when the parameters changed .. rough intuition about what we might expect to be more classical a priori, for example, would be helpful to establish.

(6) The description of the results felt overwhelming on detail in general, and it was not easy to track what the important points might be for later. Even for example, when the author says  (line 248): This information will be used later, I am not entirely sure what physically ideas I am supposed to retain from this section.

(7) I particularly lost track of the significance of the discussion in the paragraph between lines 295-306 and was perhaps therefore unable to appreciate what we learned by the of sec 4.2 in line 309

(8) The conclusions themselves seem to be flat in being statements about results having been found, but again, I wasn't clear what I necessarily took away from all of this.

Reviewer 3 Report

Comments and Suggestions for Authors

The author considered the so-called multi-particle Arnol'd cat that has been already investigated in a previous paper. Here, the author computed the time autocorrelation function of the position operator and the out of time correlation for both position and momentum.

The contents and topics addressed in the paper are definitely interesting and deserved to be studied. However, I do not see many steps forwards in this paper expected for the calculations of some correlation functions. Moreover, whenever the author finds interesting behaviours of the correlation function like Eq. (13)--a sort of anomalous behavior is observed--he neither seriously discusses the implication of these results, nor shows a comparison with the literature.

I think that this criticism can be easily addressed and therefore I would strongly suggest to the author to take care of that.

Thus, my major criticisms about the serious flaws of the paper are the exposition of the content and discussion of the results.

Exposition of the contents

Almost all of the contents of this paper are already discussed in the previous paper of the author; however, this should not preclude at all a better explanation of the model in the introductory section. This is not a letter format, therefore, a suitable background should be include summarising the important equations also in the current paper. In this way, also for defining the nomenclature one should read another paper. I do not like it at all.

There is not emphasis and distinction between what is the classical and what quantum in the sections where the correlation functions are computed. It would be better to explicitly writing and making clearer this aspect.

Discussion of the results

It is fine to present a paper with just calculation of observables but it is also important to have an understanding about the physical consequences of the results.

  1. When scattering of the large particle with the smaller ones is considered, the ACF features a faster

  2. 324  power–law decay with exponent larger than one, again for a physically acceptable range of times.

 This is well-known and this process is called anomalous behavior/diffusion. It has been observed in many systems moving in complex environment (and not only). Diffusion process indeed can manifest also complicated behavior and interestingly, it can manifest also when two interacting subsystems (whose total energy is conserved) are observed from the perspective of only one. Thus, from a subsystem's perspective the system comes to appear not at equilibrium and computing the correlation function of variables of one subsystem it is possible to find behavior like in Eq. (13).

Thus, it is in my opinion important to make a comparison and think about the consequence but also analogy with classical systems whose manifest anomalous behavior.

Minor remarks

Line 137: N the dimension of the considered Hilbert space should be specified that it is integer otherwise explain better the meaning of the sentence: "N is proportional to the mass M of the particle". This is confusing since the mass can take any (positive) real value.

Below Eq. (12). "n figure the same increase is appears to characterise—in"

rephrase better the sentence, maybe "the same growth appeared/appears ..."
